# Post-Transcriptional Modifications of Conserved Nucleotides in the T-Loop of tRNA: A Tale of Functional Convergent Evolution

**DOI:** 10.3390/genes12020140

**Published:** 2021-01-22

**Authors:** Martine Roovers, Louis Droogmans, Henri Grosjean

**Affiliations:** 1Labiris, Avenue E. Gryson 1, 1070 Bruxelles, Belgium; mroovers@spfb.brussels; 2Laboratoire de Chimie Biologique, Université Libre de Bruxelles (ULB), Labiris, Avenue E. Gryson 1, 1070 Bruxelles, Belgium; louis.droogmans@ulb.be

**Keywords:** tRNA, T-loop, nucleotide modifications, evolution

## Abstract

The high conservation of nucleotides of the T-loop, including their chemical identity, are hallmarks of tRNAs from organisms belonging to the three Domains of Life. These structural characteristics allow the T-loop to adopt a peculiar intraloop conformation able to interact specifically with other conserved residues of the D-loop, which ultimately folds the mature tRNA in a unique functional canonical L-shaped architecture. Paradoxically, despite the high conservation of modified nucleotides in the T-loop, enzymes catalyzing their formation depend mostly on the considered organism, attesting for an independent but convergent evolution of the post-transcriptional modification processes. The driving force behind this is the preservation of a native conformation of the tRNA elbow that underlies the various interactions of tRNA molecules with different cellular components.

## 1. Introduction

Transfer RNAs play a central role in deciphering the genetic code during the complex multistep translation process. To fulfil this main cellular function, each isoacceptor species has to be sufficiently distinct to serve as substrate for a large array of specific proteins and enzymes such as modification enzymes and cognate aminoacyl-tRNA synthetase (identity problem), while it also has to be sufficiently identical to serve in a uniform way for interactions with other partners such as elongation/initiation factors and with elements of the ribosome during the various steps of the translation process (conformity problem). To this end, these small RNA molecules have to fold into a uniform 3D-conformation, the so-called L-shaped architecture (reviewed in [1,2]). Conservation of certain nucleotides at key positions combined with chemical alterations of selected nucleotides catalyzed by specific enzymes during the post-transcriptional maturation processes contribute to solve both the identity and conformity problems of the tRNA repertoire within a given cell. These also allow to regulate tRNA plasticity according to the physiological conditions of the cell (especially the growth temperature) and fine-tune their interactions with the various cellular components of the translation machinery (reviewed in [3,4]). Notice, a few metazoan mitochondrial tRNAs lack the D-or T-arm and contain less modified nucleotides as the cytoplasmic counterparts. However, based on computer simulations, the overall L-shaped conformation of these “truncated” tRNAs appears to be maintained through combinatorial networks of alternative tertiary interactions (for details see [5,6]). These atypical tRNAs will not be considered in this review.

Here, we examine the structural parameters that allow fully mature cytoplasmic tRNAs from different origins (*Bacteria*, *Eukarya*, and *Archaea*) to adopt a universal L-shaped three-dimensional structure. Particularly important is the remarkable conservation of nucleotides (modified or not) at each of the 7 positions of the T-loop. A few nucleotides of the T-loop also specifically interact with a few conserved bases of the D-loop. The resulting L-shaped tRNA tertiary structure is further stabilized by additional tertiary base pairs between conserved nucleotides of the D-loop and of the V-loop. While conservation of nucleotides at given positions of the present-day tRNAs results from long coevolution of tRNA population with all elements of translation machinery [7,8], the enzymes that produce identical modified nucleotides are often distinct from one organism to another. This is particularly true when considering organisms belonging to different Domains of Life, attesting for convergent type of enzyme evolution for maintaining a universal functional tertiary tRNA fold.

## 2. The Intricate T-D-Loop Anatomy

Figure 1A displays the nearly universal secondary structure of mature tRNA sequences in which the conserved and semiconserved nucleotides present in both the T-loop and the D-loop are highlighted. In the T-loop, those nucleotides which are usually post-transcriptionally modified are indicated in red (Figure 1A,B and Figure 2). Their presence is dependent on both the type of isoacceptor tRNA and the group of organisms considered (*Bacteria*, *Eukarya*, and *Archaea*) (Figure 1C). In the D-loop, only 2′-*O*-ribose methylated G at position 18 (Gm) is mentioned among others not discussed here. The case of initiator tRNA of *Eukarya* is special because an A instead of the universally conserved U is found at position 54 of T-loop [9,10].

X-ray diffraction data of several tRNAs reveal the presence of a *trans* Watson–Crick/Hoogsteen pair between the uridine-54 and the adenosine-58 (both in *anti*-conformation) [11,12,13]. In the case of eukaryotic tRNA_i_^Met^, a longer purine–purine Hoogsteen pair (A54-A58) is found [14]. This conserved U54 (A54)-A58 base-pair stacks with the conserved G53-C61 base-pair at the end of the helical T-arm. This forces the conserved pyrimidine-60 (notified as Y) and nucleotide-59 (notified as N for any type of base, but mostly A or U) to flip out of the remaining 3-member mini-T-loop composed of (i) the conserved U55, which stacks with U54 and interacts with G18 of the D-loop, (ii) C56, which pairs with G19, and (iii) purine-57 (notified as R; mostly G in *Bacteria* and *Eukarya* but A in *Archaea*). The residue R57 is sandwiched between A58-U54 and G19-C56. Thus, the series of nucleotides C61, A58, G18, R57, and C56-G19 are forming a continuous stack (see right part of Figure 1A and Figure 3). The flipped dinucleotide, located between the T- and the D-domain is further stabilized out of the rest of T-loop by an H-bond between the 5′-phosphate of pyrimidine 60 and N^4^ of C61. This explains the strict conservation of the G53-C61 base pair at the end of the T-stem. In this intricate network of interacting nucleotides, key contacts are the U54-A58 Hoogsteen pair (highlighted in red in Figure 3) and the base pairs formed between the conserved U55 and C56 of the T-loop and the conserved G18 and G19 of the D-loop (highlighted in black in Figure 3). The region where the D-loop and the T-loop interact is designated as the tRNA elbow [15]. The final functional L-shaped conformation is further stabilized by a few additional tertiary base pairs between other conserved or semiconserved nucleotides of the D-loop and V-loop, such as the conserved so-called Levitt base-pair R15-Y48 [15,16] and the conserved triple interacting bases R21-A14-U8 [2,17]. Microstructural heterogeneities in the hinge part of the 3D-core results mostly from the variable number of additional bases present in the D-loop (often dihydrouridine) and V-loop of the different isoacceptor tRNAs (indicated by small blue triangles in Figure 1A). This variability in the hinge region allows some tRNA flexibility but does not alter much the global canonical L-shaped tRNA architecture that is mostly dominated by the T-D-loop kissing-type of complex (elbow).

**Figure 1 genes-12-00140-f001:**
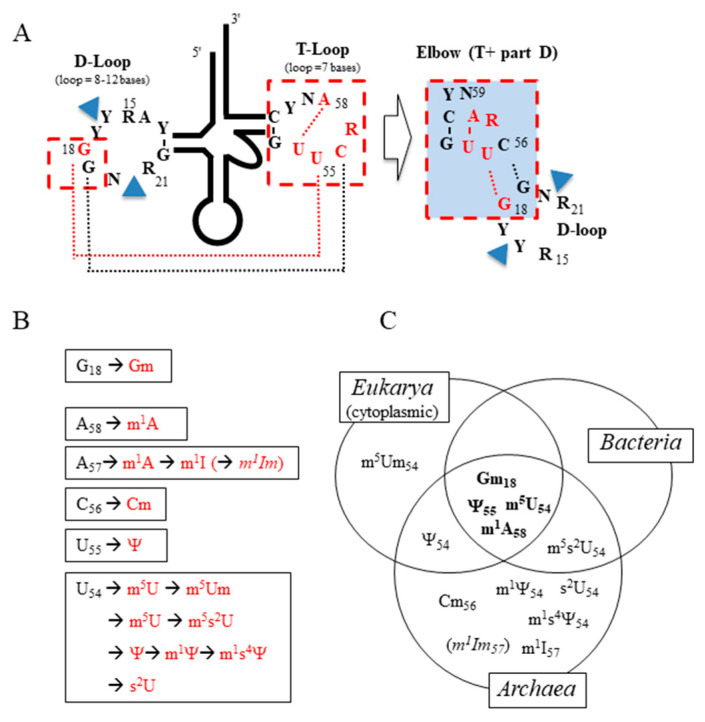
Presence of modified nucleotides in the tRNA elbow structure. (**A**) Two-dimensional representation of tRNA emphasizing conserved residues involved in D-and T-loop interaction. In red are the residues subject to enzymatic modifications. The little blue box on the right corresponds to the one in Figure 3 (see text for explanation). R stands for purine, Y for pyrimidine, and N for any of the four canonical nucleotides. (**B**) Residues at G18 and in the T-loop and their observed modifications are shown in red. (**C**) Distribution of modified residues of the T-loop among organisms of the three Domains of Life. Acronyms for modified nucleotides are those used in Modomics [10]. The 2′-*O*-methylation of m^1^I57 into m^1^Im57 is putative (see text).

**Figure 2 genes-12-00140-f002:**
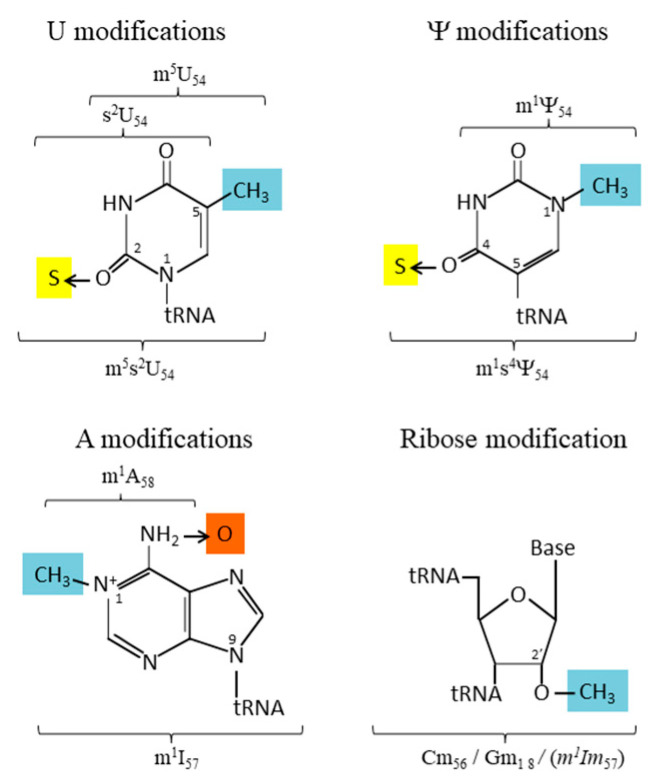
Chemical structure of the modified nucleotides found in the T-loop and G18.

**Figure 3 genes-12-00140-f003:**
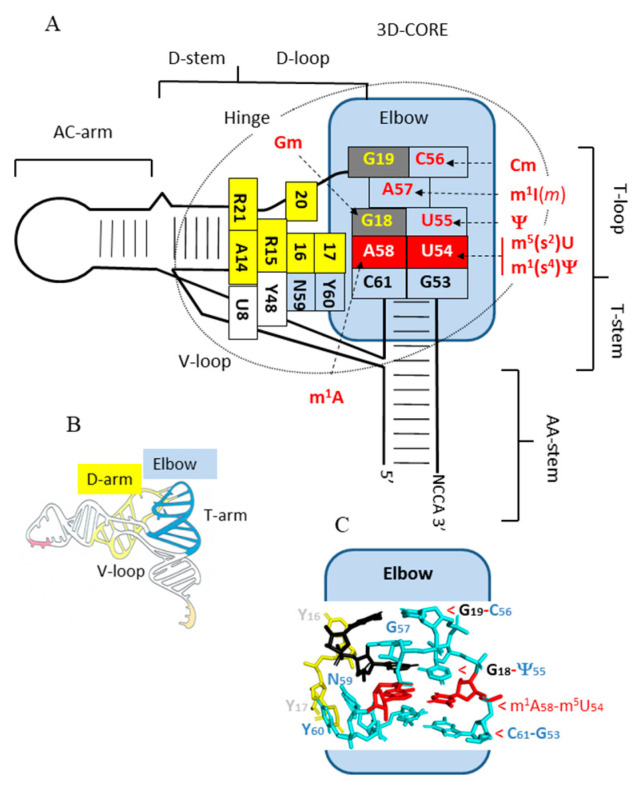
Scheme recapitulating the organization of the elbow within the general 3D organization of tRNA. (**A**) Rectangles correspond to nucleotides involved in stacking or base-pairing with another nucleotide of the T- and D-loop. Residues involved in the *trans* Watson–Crick/Hoogsteen interaction are in red boxes, other residues of the T-loop are in blue boxes, while those of the D-loop are in yellow boxes, except G18 and G19 in black boxes. The possible modifications of each residue are indicated in red. (**B**) Schematic representation of the 3D architecture of tRNA molecule. (**C**) The 3D structure of tRNA elbow as found in the crystal structure at 1.93 Å resolution of yeast tRNA^Phe^ harboring m^5^U54, Ψ55, and m^1^A58 [13]. The *trans* Watson–Crick/Hoogsteen interaction is in red, while G18 and G19 are in black.

## 3. Modified Nucleotides within the T-Loop and G18 Stabilize the Elbow of tRNA

The chemical structures of the various modified nucleotides found in the T-loop are depicted in Figure 2. Isomerization of U55 into the universally conserved Ψ55 favors the 3′-*endo* sugar pucker as in the A-form of RNA and enhances the stacking interaction with the *trans* Watson–Crick/Hoogsteen pair U54-A58 and the nearest-neighbor purine 57 [18]. The additional H-bound at N^1^ of Ψ55 involves the 5′-adjacent phosphate via a coordinated water molecule [19]. Likewise, methylation of the C^5^ atom of U54 into m^5^U and isomerization of U54 into Ψ54 followed by the N^1^-methylation of Ψ54 similarly favors the 3′-*endo* ribose puckering of the modified nucleotides and reinforces the hydrophobicity of the base. This increases base stacking with the neighboring bases G53 and Ψ55. Methyl groups also increase steric hindrance, block hydrogen bonds at Watson–Crick positions, and increase resistance against hydrolysis by nucleases. In thermophilic *Bacteria* and *Archaea* (growing at temperatures above 60 °C), the uridine-54 is almost invariably hypermodified into thiolated derivatives (m^5^s^2^U or m^1^s^4^Ψ) [20,21]. Remarkably, hypermodified m^5^s^2^U and m^1^s^4^Ψ at position 54 are isosteric, which implies that in both cases, the bulky highly polarizable thio group reinforces the 3′-*endo* ribose puckering of the nucleotide. This considerably enhances the stacking interaction with the nearest-neighboring nucleotides G53 and Ψ55 [22] and allows base pairing in an identical way with the *trans*-Hoogsteen nucleotide 58.

Methylation of the 2′-hydroxyl of the ribose favors the 3′-*endo*-conformation and changes the hydration sphere around the oxygen of the ribose [23]. Further, 2′-*O*-ribose methylation of cytidine at position 56 (Cm) was found in tRNAs of the crenarchaeon *Pyrobaculum aerophilum* [24] and also of N^1^-methyl-inosine at position 57 (m^1^Im) in tRNAs of *Pyrobaculum islandicum* and of *Pyrodictium occultum*. However, in this latter case information comes from analysis of bulk tRNA with no evidence for the presence of m^1^Im at position 57 exclusively [25]. 2′-*O*-ribose methylation of m^5^U at position 54 (m^5^Um) exists in certain tRNAs of *Eukarya* [26]. Residue Gm18 in the D-loop is found in tRNAs of all Domains of Life but are more frequently found in thermophilic *Bacteria* and hyperthermophilic *Archaea* [20]. This methylation is diagonally opposite to the 2′-*O*-methylation of C56 (Cm) that base pairs with the conserved neighboring G19.

Methylation of the N^1^ atom of adenosine 58 blocks a Watson–Crick position and favors the formation of an intraloop Watson–Crick/Hoogsteen pairing with m^5^U54 or Ψ/m^1^Ψ54. The electropositive character of this methylated adenosine also promotes ionic interaction with the negatively neighboring charged phosphates of the backbone. In *Bacteria* and *Eukarya*, the conserved purine 57 is more often G57 than A57, while in thermophilic and hyperthermophilic *Archaea*, a neutral N^1^-methylinosine (m^1^I) is often found instead. Such deaminated adenosine can establish H-bonds as a guanosine. Nucleotide in position 57 (G57 or derivatives of A57) is sandwiched between the base pairs G18-U55 and G19-C56 of the tRNA elbow (Figure 3). Notice, the presence of m^1^A58 and/or m^1^I57 depends much on the archaeon considered and the type of tRNA analyzed (especially the nature of nucleotide-59, see below). For example, most tRNAs from halophilic *Archaea* such as *Haloferax volcanii* display m^1^I57 and unmodified A58 [27,28], while tRNAs of thermophilic and hyperthermophilic *Archaea* often display m^1^A58 and/or m^1^I57 [29] and references therein.

In sum, the highly conserved nucleotide modifications of the T-loop and of G18 of the D-loop mainly serve to enhance the intrinsic stability of the tRNA elbow and also to enhance the affinity between the T- and D-loop, especially in the thermophilic organisms [30].

## 4. The Enzymatic Toolbox for Nucleotide Modifications at Positions 18 (D-Loop) and 54–58 (T-Loop) of tRNA

The elbow region of tRNAs of *Bacteria* and *Eukarya* contains characteristic conserved noncanonical nucleotides at positions 18, 54, 55, and 58, while in *Archaea*, additional conserved modifications also occur at positions 56 and 57 (Figure 1 and Figure 2). They are all catalyzed by specific enzymes during the tRNA maturation process (Figure 4 and Figure 5).

### 4.1. G18 Modification in the D-Loop

In tRNA species of most Gram-negative *Bacteria* the highly conserved guanosine-18 is converted to a 2′-*O*-methylated derivative (Gm). Gm18 is absent in Gram-positive *Bacteria* with a few exceptions such as the thermophilic *Geobacillus* sp (reviewed in [31]). This modification is accomplished by the dimeric S-adenosyl-L-methionine (SAM)-dependent methyltransferase (MTase) TrmH belonging to the SPOUT superfamily [32] (COG0566–Cluster of Orthologous Genes, see [33]). Two types of TrmH enzymes exist with different tRNA recognition specificities: one methylating all tRNA types, like TrmH of *Thermus thermophilus*, and the other one methylating only a specific subset of tRNAs like TrmH of *Escherichia coli* and *Aquifex aeolicus* [34,35,36], reviewed in [37].

In yeast, Gm18 formation is catalyzed by the dimeric TRM3 MTase also belonging to the SPOUT superfamily (COG0566). Yeast TRM3 exhibits a very long N-terminal extension of unknown function [38]. The human homologous enzyme is TARBP1 [39].

In *Archaea*, while Gm18 has been experimentally identified in tRNA of different thermophilic organisms [29], no stand-alone homolog of TrmH/TRM3 has yet been identified. Instead, an alternative pathway involving a C/D box guide has been suggested, as in the case of 2′-*O*-ribose methylation of C56 of the T-loop (see below) and also for other nucleotides in the anticodon loop and position 50 of the T-arm [40,41]. Methylation will then occur via a nucleoprotein complex involving the ad hoc C/D box guide RNA, two protein cofactors Nop5 and L7Ae, and a SAM-dependent fibrillarin-like MTase (FlpA, belonging to COG1889).

**Figure 4 genes-12-00140-f004:**
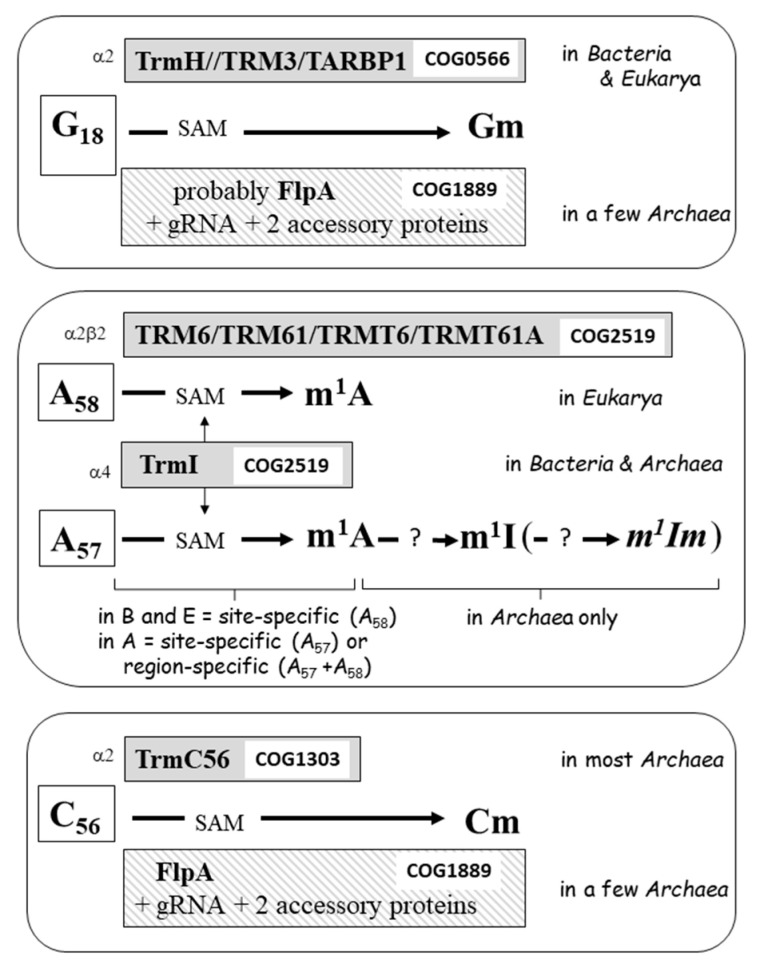
Enzymatic systems involved in post-transcriptional modifications of selected nucleotides of the D-and T-loop of tRNAs of various organisms. Acronyms of enzymes are boxed with various backgrounds according to the type of modification together with the COG (Cluster of Orthologous Genes) number. The Greek symbols indicate the oligomerization state of the functional enzymes. Only in a few cases, the enzyme corresponding to a given modification remains unknown (indicated by?). The 2′-*O*-methylation of m^1^I57 into m^1^Im57 is putative (see text). Acronyms for modification enzymes are those used in Modomics [10], except for those in human [39]. gRNA stands for guide RNA. A, B, and E refer, respectively, to *Archaea*, *Bacteria*, and *Eukarya*.

### 4.2. Modification of A58 and A57

N^1^-methyladenosine (m^1^A) is often present at position 58 of tRNAs of organisms belonging to all three Domains of Life. Its formation, role, and structural implications has been the subject of various reviews [42,43,44,45,46].

In *Bacteria*, like *T. thermophilus*, for which m^1^A58 is present in all sequenced tRNAs, the enzyme catalyzing this N^1^-adenine methylation is the homotetrameric TrmI [47,48]. This SAM-dependent enzyme belongs to the Rossman fold-like (RFM/class I) family of MTases (COG2519) and is site-specific for the N^1^-position of A58 [47]. The T-loop in combination with the aminoacyl-stem and the variable region are required for efficient methyl transfer [49].

In *Eukarya*, like yeast and human, a site-specific A58 TrmI-like ortholog exists but displays a different protein architecture. Here, the enzyme is a heterotetramer composed of two different kinds of subunits: a catalytic SAM-binding protein (TRM6/TRMT6 belonging to RFM/class I) and an evolutionary related RNA binding protein (TRM61/TRMT61A), both belonging to COG2519. As for bacterial TrmI, eukaryal N^1^-adenine MTase is site-specific for A58 and requires the T-loop in combination with the aminoacyl-stem [50,51,52,53], reviewed in [54].

In hyperthermophilic *Archaea*, like *Pyrococcus furiosus* or *Pyrococcus abyssi*, the TrmI ortholog works as a homotetramer. However, at variance with the bacterial enzyme, it involves intersubunit disulfide bridges that reinforce the tetrameric oligomerization [55,56]. This archaeal TrmI can work in vitro on a minisubstrate composed of only the T-branch [50]. Remarkably, it methylates the N^1^-atom of both neighboring adenosines 58 and 57 of the T-loop, with a preference for position A57. This region specificity depends on the presence of an adenosine at position 59 of tRNA. In absence of A59, archaeal TrmI behaves as an A57-specific MTase [57]. A model based on fluorescence measurements of 2-aminopurine placed at positions 57 or 58, showed that methylation is sequential, forming m^1^A57 before m^1^A58 [58]. In archaeal tRNAs, m^1^A57 never accumulates, i.e., the 6-amino group of adenine is always very efficiently deaminated by a new type of hydrolytic deaminase (still to be identified) leading to formation of 1-methylinosine (m^1^I57) [59,60]. The nucleotide 57 is sandwiched between G19-C56 and G18-U55 (see above). The still unknown m^1^A-dependent deaminase removes the positive electrostatic charge of the N^1^-atom of m^1^A (Figure 2), leading to the neutral m^1^I residue. Although it is difficult to evaluate the molecular effects of the latter modification, it is conceivable that m^1^I stabilizes stacking interactions better than m^1^A. On the contrary, the presence of a positively charged m^1^A58 has been demonstrated to be critical for the folding of the T-loop and proper functioning of certain tRNAs (discussed in [61,62]). In the majority of other bacterial and eukaryal tRNAs, nucleotide 57 is usually G or unmodified A. As for N^6^-methyladenosine (m^6^A) in mRNA, m^1^A in human tRNAs has been found to be occasionally demethylated by a specific enzyme [63], reviewed in [64].

### 4.3. Modification of C56

The presence of 2′-*O*-methylcytidine (Cm) at position 56 is a hallmark of archaeal tRNAs. It has never been found in any tRNA of *Bacteria* nor of *Eukarya*. As for the formation of Gm18 (see above), two different evolutionary unrelated enzymatic mechanisms exist to catalyze this ribose methylation. In most *Archaea*, Cm56 formation is achieved by the stand-alone SAM-dependent Trm56 belonging to COG1303 [65]. Little is known about tRNA recognition by this enzyme. Nevertheless it seems that besides the C56, the enzyme recognizes more than just the T-stem-loop, at least in vitro and with unmodified tRNA transcripts as substrates [65,66]. This dimeric MTase belongs to the same SPOUT superfamily as the G18-MTase (TrmH/TRM3/TARBP1) but belongs to a different COG [37,67]. In other *Archaea*, such as the hyperthermophilic *Pyrobaculum aerophilum*, Cm56 formation is achieved by a nucleoprotein complex composed of a fibrillarin-like MTase (FlpA, COG1889), two protein cofactors (Nop5 and L7Ae) and a specific small C/D guide RNA that allows the FlpA enzyme to find its target nucleotide [65], reviewed in [24]. This enzymatic pathway is the same as the one catalyzing formation of Gm18, except that distinct guide RNAs are used. Interestingly, a simplified heterodimer composed of *P. abyssi* FlpA and Nop5 can perform SAM-dependent 2′-*O*-methylation of RNA in vitro independently of L7Ae and C/D guide RNA. This stand-alone activity of dimeric FlpA-Nop5 may reflect the ancestral activity of the enzyme, before its recruitment into larger complexes [68].

### 4.4. Modification of U55

Pseudouridine (Ψ) at position 55 is the most universally conserved modified nucleotide in tRNA of all three phylogenetic Domains of Life. Nevertheless, as for several other modified nucleotides discussed in this work, different enzymatic strategies are used to catalyze this simple isomerization reaction in different organisms.

In *Bacteria* and *Eukarya*, stand-alone Ψ55-synthase TruB (in *E. coli*) and its ortholog PUS4/TRUB1 (in *Saccharomyces cerevisiae*/human) catalyze this modification [69,70,71]. These monomeric enzymes belong to COG0130, which also includes the close homolog Cbf5 (see below). Bacterial TruB in complex with its RNA substrate is probably the best characterized system involving a modification enzyme [72,73,74,75]. This bacterial TruB enzyme, as well as its eukaryal orthologs PUS4/TRUB1 require no cofactors and accept as substrate a simple 17 base oligonucleotide analog to the T-arm and containing the sequence 53-GUUYRANYC-61 [70,76]. This explains why these enzymes efficiently work on unmodified precursor tRNAs (including synthetic molecules) and on many cellular RNAs (other than tRNAs) in vitro and in vivo, as long as the target position in the RNA substrate remains accessible to the enzyme. Besides its catalytic function, bacterial TruB also acts as RNA chaperone [77].

In *Eukarya* and *Archaea*, but not in *Bacteria*, two additional enzymes are implicated in Ψ55 formation. The first usually functions in a ribonucleoprotein complex that utilizes a well-structured box H/ACA guide RNA to target the sequence to be modified, two or three protein cofactors (Nop10, Gar1, and Nhp2 in *Eukarya* and Nop10, Gar1, and L7Ae in *Archaea*) and the enzyme Cbf5 belonging to the same COG0130 as PUS4/TRUB1. In *Eukarya*, such H/ACA guided enzymatic system is used essentially for site-specific catalysis of Ψ formation in various cellular RNA targets, mostly rRNA, and not tRNA (reviewed in [78,79,80]). In *Archaea*, Cbf5 can function as a stand-alone Ψ-synthase for tRNA Ψ55 formation, at least in vitro [81,82,83,84]. This guide-independent tRNA specific activity of archaeal Cbf5 is enhanced in vitro by the proteins Nop10 and Gar1, probably by stabilizing the active conformation of Cbf5 [82,85,86,87]. It is worth mentioning that the *Haloferax volcanii cbf5* deletion strain still contains tRNA carrying Ψ55, attesting that Cbf5 is not the main physiological Ψ55 synthase, at least in this archaeon [88]. This is another example of the probable ancestral activity retained in the present-day Cbf5 enzyme that was recruited into larger complexes later during evolution.

In *Archaea*, the genuine enzyme catalyzing Ψ55 formation in tRNA is the stand-alone Pus10 belonging to COG1258, phylogenetically unrelated to TruB/PUS4/TRUB1 (COG0130) [81,83]. This monomeric Ψ-synthase needs no protein cofactors and is present in a large variety of *Archaea* (mostly in *Euryarchaeota* and a few *Crenarchaeota*). It is also present in a few *Eukarya* (including human) but not in *Bacteria* [89,90]. In archaeal tRNAs the main target is U55, but depending on the archaeon considered, Pus10 behaves as a region-specific enzyme catalyzing Ψ formation at both positions 55 and 54 (see below).

In human, the situation is more complicated due to the coexistence of nuclear/cytoplasmic Ψ55-specific TRUB1 and of mitochondrial TRUB2 (not discussed here). While recombinant human PUS10 can generate Ψ55 in human cytoplasmic tRNAs in vitro [91], its main cellular function is to promote pseudouridylation of U54 in only a small subset of the total cytoplasmic tRNA population (see below), the formation of Ψ55 in the majority of the other human tRNAs being achieved in the nucleus by TRUB1.

**Figure 5 genes-12-00140-f005:**
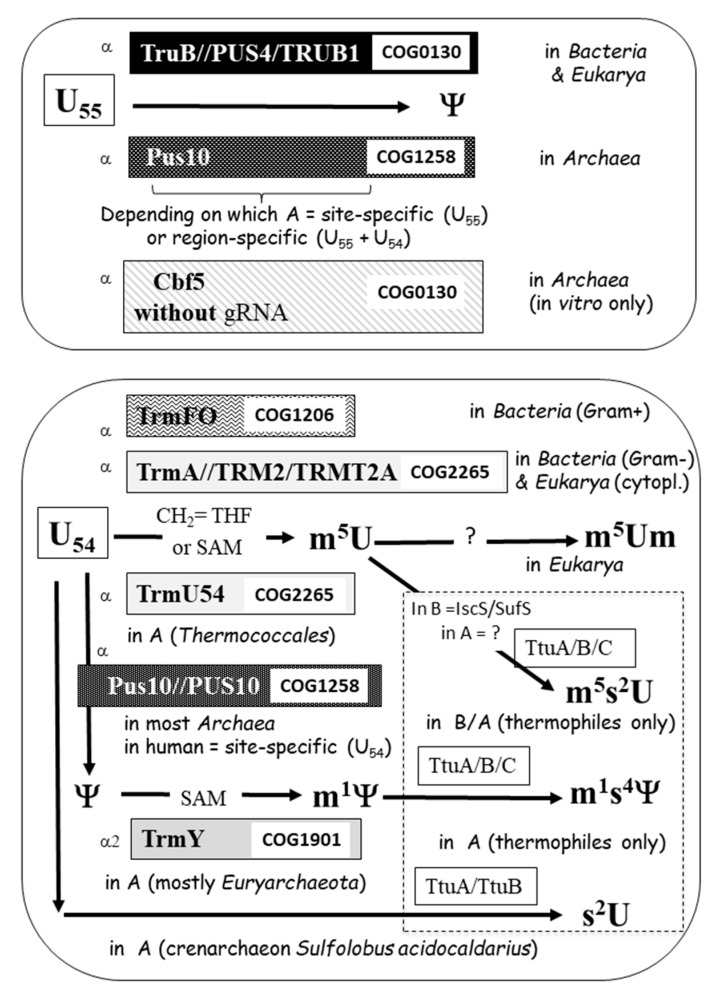
Enzymatic systems involved in post-transcriptional modifications of U55 and U54. Same as legend to Figure 4.

### 4.5. Modification of U54

Methylation of the C^5^-atom of uridine at position 54 (m^5^U, also referred as ribothymidine rT), is the second most commonly found modified nucleotide in tRNAs of *Bacteria* and *Eukarya*. In *Archaea*, it is only found in a small subset of organisms belonging to *Thermococcales* and the sister lineage *Nanoarchaeota*.

In *E. coli*, m^5^U54 formation is catalyzed by the monomeric SAM-dependent TrmA enzyme of the RFM superfamily belonging to GOG2265 [92]. Its presence, but not its enzymatic activity is essential for *E. coli* viability [93,94]. This is probably explained by the chaperone activity of TrmA during tRNA maturation [95]. In yeast, the orthologous enzyme is TRM2 and in human TRMT2A [96,97].

In *Bacteria*, the use of SAM as a methyl donor by the TrmA enzyme is common for m^5^U54 formation in β-γ-and ε-*Proteobacteria*, while in *Firmicutes*, α-and δ-*P**roteobacteria* as well as in *Cyanobacteria* and *Deinococci*, the enzyme responsible for the methylation of U54 is the flavoprotein TrmFO. This enzyme uses N^5^, N^10^-methylenetetrahydrofolate (CH_2_ = THF) in a more complex mechanism than the rather simple methyl transfer from SAM. Here, a methylene group is transferred first and a subsequent reduction step involving the flavin adenine dinucleotide hydroquinone (FADH^-^) is required to form the final methylated derivative [98,99], reviewed in [100]. The monomeric folate/FAD-dependent TrmFO enzyme belongs to COG1206. In *Mollicutes*, a group derived from *Firmicutes* and displaying extensive genome erosion (retrograde evolution), several SAM-dependent m^5^U MTases present in their common ancestor have been lost and replaced by folate-dependent paralogs [101,102]. These are clear examples of independent and mutually exclusive evolution of analogous enzymes catalyzing the formation of the same modification (m^5^U54) using totally unrelated mechanisms in different bacterial lineages.

In the hyperthermophilic *Archaea* of the phyla *Thermococci* (such as *P. abyssi* and *P. furiosus*) and *Nanoarchaeota* (as *Nanoarchaeum equitans*), m^5^U54 formation is catalyzed by TrmU54, a SAM-dependent MTase belonging to COG2265. Although this monomeric protein belongs to the same COG as TrmA/TRM2/TRMT2A, the protein sequence of archaeal TrmU54, as well as eukaryal TRM2, are more similar to the bacterial paralog RlmD (formerly, RumA/RumT) catalyzing m^5^U1939 formation in 23S rRNA [103]. One main difference is the presence of an iron–sulfur cluster in the archaeal TrmU54 [104]. In both cases, these two enzymes, TrmU54 and TRM2, arose from an ancestral horizontal transfer (endosymbiosis in the case of TRM2) followed by a shift of function from rRNA to tRNA modification [103,105].

All these MTases are site-specific and methylate exclusively the C^5^-atom of U54. As for TruB/PUS4 above, the minimal RNA structure for substrate recognition by TrmA, TRM2, and TrmU54 is a 5-base-pair stem, eventually prolonged with the amino acid arm, and a seven-membered loop as in a canonical T-loop of tRNA [50,66,70,75,106,107,108]. The same applies to tRNA recognition by bacterial TrmFO [109].

Depending on the organism, residue m^5^U54 can be further modified by either 2′-*O*-methylation or thiolation. In rabbit and human, 2′-*O*-methylated derivative of m^5^U54 (m^5^Um) was reported for tRNA^Lys3^ [26,110]. The corresponding MTase remains to be identified. In both thermophilic *Bacteria* and *Archaea*, m^5^U54 is further modified in m^5^s^2^U54 by a thiolation enzymatic machinery involving several proteins (TtuA, TtuB, TtuC, and cysteine desulfurases IscS or SufS in *Bacteria*, not yet known in *Archaea*) [111], reviewed in [112]. In *T. thermophilus*, the level of thiolation enzyme and, consequently, the degree of tRNA m^5^U thiolation increases with increasing cultivation temperatures [113]. The major determinants for 2-thiolation of m^5^U54 is the presence of the conserved C56 and m^1^A58 of the T-loop. Tertiary interactions between the T- and D-loops and nonconserved nucleotides in the T-loop are not important for the reaction [114]. The importance of m^1^A58 for m^5^s^2^U54 formation is supported by the fact that a Δ *trmI* strain of *T. thermophilus* loses its ability to grow at higher temperature range [47,113], reviewed in [20]. Moreover, the presence of m^7^G46 in the V-loop allows a better efficient thiolation reaction attesting for a clear interplay between the different modifications in tRNA [115].

In *Archaea*, other than *Thermococci* and *Nanoarchaeota*, and a few *Eukarya*, U54 is modified to Ψ. This isomerization reaction is catalyzed by the Ψ-synthase Pus10/PUS10 already mentioned above for catalyzing Ψ55 formation. However, depending on the archaeon, Pus10 can be either site-specific for U54 or region-specific for both U55 and U54. When tested in vitro, recombinant Pus10 of *Methanocaldococcus jannaschii* catalyzes formation of both Ψ55 and Ψ54 with the same efficiency, whereas recombinant Pus10 of *P. furiosus* is inefficient for catalyzing Ψ54 formation under the same experimental conditions [83]. When tested in vivo, recombinant *M. jannaschii* Pus10 also efficiently forms both Ψ55 and Ψ54 in tRNAs of a double (*truB-trmA*)-mutant *E. coli* strain. In contrast, recombinant *P. furiosus* Pus10 only forms Ψ55 in the same tRNAs of the *E. coli* double-mutant strain [116]. Such versatility of archaeal Pus10 is consistent with the fact that in naturally occurring tRNAs of *P. furiosus* an m^5^U54 is present, while in *M. jannaschii*, tRNAs m^1^Ψ54 is found. Thus, the restricted specificity of *P. furiosus* Pus10 for U55 avoids a competition with methylation of U54 by TrmA [83]. Recombinant human PUS10 can also catalyze Ψ55 and Ψ54 formation in vitro [117], attesting for adaptation of the enzyme specificity according to the cellular need [116]. Maximal Ψ54 synthase activity is observed for tRNA substrates with the sequence 53-GUUCAm^1^AAUC-61 along with a stable acceptor stem. As for the archaeal Pus10, the crystal structure of human PUS10 shows the presence of a conserved Ψ-synthase catalytic domain fused to a characteristic THUMP domain [118]. THUMP is an RNA-binding module that is also present in a few other tRNA modification enzymes such as thiouridine synthases and MTases [119,120,121,122,123,124].

In *Archaea* (mostly *Euryarchaeota* but also in a few *Crenarchaeota*), Ψ54 can be further methylated into m^1^Ψ54 by a SAM-dependent MTase (TrmY) belonging to COG1901 [125,126]. The structure of this dimeric enzyme shows that it belongs to the SPOUT superfamily [127]. In Vitro experiments with purified recombinant region-specific *M. jannaschii* Pus10 and TrmY demonstrates that TrmY (as Pus10 above) can produce m^1^Ψ in a 17-base fragment of a tRNA transcript corresponding to the T-arm of which U55 (possibly Ψ55) cannot be mutated [125]. In thermophilic *Archaea*, residue m^1^Ψ54 can be further thiolated to m^1^s^4^Ψ54 most probably by the same thiolation enzymatic machinery reported above for m^5^s^2^U54 formation in thermophilic *Bacteria* (involving proteins TtuA, TtuB, TtuC, and cysteine desulfurases not yet identified in *Archaea*). For example, in tRNAs of *Ignicoccus hospitalis*, a member of the *Crenarchaeota*, m^1^s^4^Ψ54, is found, whereas in tRNAs of *Nanoarchaeum equitans*, closely related to the *Thermococcales*, m^5^s^2^U54 is present [21]. These two *Archaea* are growing physically attached as obligate partners exchanging various metabolites under hyperthermophilic conditions [128]. Remarkably, the methyl and thio groups in m^1^s^4^Ψ and m^5^s^2^U are presented in the same spatial manner for the stabilization of the 3′ *endo*-*anti* conformation of nucleotide 54, showing that these *Archaea* use different but structurally equivalent modifications to stabilize their T-loops. This illustrates how different solutions have been adopted during evolution to fulfil the same function.

Finally, in the case of the hyperthermophilic crenarchaeon *Sulfolobus acidocaldarius*, s^2^U has been found at position 54 of several tRNAs. In the particular case of tRNA^Val^, a methylthio U-derivative was also found, but its chemical structure remains elusive [29]. Examination of the *S. acidocaldarius* genome reveals the presence of genes coding for TtuA and TtuB responsible for thiolation of U54 but no gene coding for TrmA (catalyzing m^5^U54), Pus10 (catalyzing Ψ54 in addition of Ψ55), and TrmY (catalyzing m^1^Ψ54) was found. One possibility is that ribose methylation of U54 occurs in only a few tRNA species, such as the initiator tRNA [129]. Indeed, 2′-*O*-methylated nucleotides are found at several positions of various *S. acidocaldarius* tRNAs and are known to be formed by the guide RNA-dependent FlpA machinery [130,131]. However, no guide RNA specific for U54 methylation has been reported up to now. Therefore, in absence of a clear-cut evidence for Um54 existence and of convincing information about the corresponding methylation system, this problem remains unsolved.

## 5. Temporal Order of Nucleotide Modifications and Their Interdependencies

After transcription, each tRNA primary transcript (pre-tRNA) is engaged in a complex multistep maturation during which 5′-leader and 3′-trailer are removed, introns are spliced, 5′-G (in the case of tRNA^His^) and CCA-end are added (if not encoded in the genome), and a bunch of nucleotides become modified. In *Eukarya*, this process starts in the nucleus and terminates in the cytoplasm. Along this complex sequential maturation, both the structural conformation and the sequence of the pre-tRNAs are progressively transformed in fully mature and functional tRNA molecules with the canonical L-shaped structure.

During this process, enzymatic modifications introduced by a machinery utilizing guide RNAs that base pair with the pre-tRNA substrate and by enzymes requiring limited small substructures are expected to occur first, while enzymes requiring an intact tRNA architecture are expected to act later. Hence, the earliest expected modifications are Gm18 and Cm56 (in *Archaea*), which both depend on guide RNA, and Ψ55, Ψ54, m^5^U, m^1^Ψ54, which all depend on stand-alone enzymes working on minimal T-arm substructures. These are precisely the modifications that were experimentally shown to be the first ones to appear during pre-tRNA processing using either in vivo or in vitro assays [50,132,133,134]. Recently, experiments aiming at monitoring the time-resolved introduction of modifications in yeast tRNA^Phe^ in vitro not only confirm these observations but also demonstrate the existence of a robust modification circuit (interdependency) starting with the introduction of Ψ55, then m^5^U54, and finally m^1^A58 [135]. Since the presence of N^1^-methyl group on A58 is required for efficient enzymatic transfer of sulfur on the C^2^-atom of m^5^U54 [113], the biogenesis of m^5^s^2^U54 by TtuA/TtuB/TtuC at elevated temperature in thermophilic *Bacteria* and possibly of m^1^s^4^Ψ in hyperthermophilic *Archaea* should occur later during tRNA maturation. These observations are in complete agreement with the theory of interdependency between the various enzymatic modification systems acting on nucleotides in the tRNA elbow, in which Ψ55-synthase TruB/PUS4/TRUB1 (and probably Pus10) appears to be the conductor, besides acting itself as a chaperone protein [136,137,138,139]. Interdependency between different nucleotide modifications and their modification pathways has been found to be particularly important for modifications in the anticodon-stem loop [60,140,141].

## 6. tRNA T-Loop Modification Enzymes Can Modify Other T-Loop Containing RNAs

Conserved base-pairings between the T-and D-loop are intrinsic properties of the tRNA itself: they are present in synthetic tRNA transcripts lacking any modified nucleotide, yet in slightly more relaxed conformation [142,143]. Genuine T-loop architecture also exists in local regions of the ribosomal RNAs [144], at the 3′ end of the plant viral TYMV RNA [145] and in bacterial tmRNA [146]. In these two last examples, TrmA/TRM2/TRMT2A was shown to catalyze m^5^U formation at the position equivalent to U54 [147,148]. The main role of post-transcriptional modifications of the various conserved nucleotides of the T-loop is therefore to fine-tune the dynamics (rigidity/flexibility) of the self-formation of this important part of the tRNA molecule and to facilitate its interaction with the D-loop. As illustrated in Figure 3, the various conserved modified nucleotides (Gm18, m^5^(s^2^)U/m^1^s^4^Ψ54, Ψ55, Cm56, and m^1^I57/m^1^A58) are very close together in the L-shaped conformation of tRNA.

## 7. Conclusions and Future Prospects: Unraveling the Evolutionary Origin of tRNA Modification Machinery

The importance of a few invariant and semi-invariant nucleotides for self-forming 3D-conformation of the T-loop of tRNAs is now well documented. This unique structural domain is characterized by a *trans* Watson–Crick/Hoogsteen pair between U54 and A58, which, in turn, allows U54, U55, the bulged pyrimidine 60, and nucleotide 59 to further interact specifically with nucleotides of the D-loop (Figure 3) [149,150]. Since transcription proceeds in the 5′ to 3′ direction, this T-loop/D-loop interaction cannot take place before almost the full sequence of the tRNA gene is transcribed. In addition, among the many modified nucleotides found in fully matured tRNA molecules, Gm18, Ψ55, m^5^U54/m^1^Ψ, m^1^I57/m^1^A58, and Cm56 are likely the first to occur in the maturation pathway, the most universally conserved Ψ55 being probably the very earliest one. These combinatorial networks of base pairings and nucleotide modifications, together with other tertiary interactions and base modifications within the entire core of the tRNA molecule, permit each individual fully matured isoacceptor tRNA species to adopt a stable still flexible L-shaped conformation [151]. In this final functional conformation, the distance between the anticodon and the CCA-3′-terminus at the opposite is around 80 Å. This property is essential for each aminoacyl-tRNA to accommodate in the decoding center of the small ribosomal subunit and then in the peptidyl-transferase center of the large subunit.

Because of the highly conserved characters of the modified nucleotides found in the T-loop and around G18-G19 among tRNAs of organisms belonging to the three Domains of Life, the simplest ones like Ψ, Cm, m^5^U, and m^1^A might have been present in the prebiotic soup [152,153,154]. The corresponding modification enzymes might have appeared later, yet very early in evolution, at least at the root of the common ancestral lineages of *Bacteria*/*Archaea* [155,156]. If the present-day enzymes would have been vertically inherited from these earliest forms, their sequences should retain substantial similarities among orthologous enzymes present in majority if not all organisms of the Tree of Life. On the contrary, and this is the main message of the present review, almost all the enzymes involved in the modification of nucleotides at positions 18, 54, 55, 56, and 57/58 differ much from one group of organisms to another, U54 being the most diversely modified one. These nonhomologous enzymes (also named “analogous enzymes” [157]) belong to different COGs, use sometimes different cofactors (such as SAM or CH2 = THF) and mechanisms (standalone or guided-RNA machineries), and recognize different portions of the tRNA molecule. They can be site-, region-, or even dual-specific (for tRNA and other types of RNA). Evidently, emergence and evolution of a given enzymatic activity is more complex than just by gene duplication and neofunctionalization followed by eventual elimination of the ancestral gene. Indeed, a gene present in a given organism can be lost and replaced by a nonorthologous paralog coding for a protein with similar or identical function (see, e.g., [102]). In addition, any time along the cellular evolution, a new enzymatic activity, or a new enzymatic mechanism can be inherited from a lateral gene transfer from an organism living in endosymbiosis or in symbiosis within the same environment. Lateral gene transfer between *Bacteria* and *Archaea* or between *Prokaryotae* and *Eukarya* are frequent [158,159]. In all cases, the driving force is to optimize an ad hoc function required by the novel needs of the cell and according to the evolving cellular physiology and/or of growth conditions. In the particular case of tRNA modification, the need is to constantly adapt and preserve the stability of the overall architecture and consequently the functionality of the tRNA molecules according to the growing conditions as well as the physiological changes that occur all along the complex evolution process.

To conclude, this review shows the complexity of tRNA modification enzymes evolution, which makes genome annotation a difficult task [160,161] and prompts to start studies on the emergence and evolution of enzymes catalyzing the formation of a given modified nucleotide at a given conserved position of the tRNA molecule.

## Data Availability

Not applicable.

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
