# Peer review of "Post-Transcriptional Modifications of Conserved Nucleotides in the T-Loop of tRNA: A Tale of Functional Convergent Evolution"

_genes, 2021, doi:10.3390/genes12020140_

Round 1

Reviewer 1 Report

Comments to the authors:

This (genes-1074881) is an interesting review.  There are many reviews that summarize relationships between modified nucleotides in tRNA and tRNA structure.  However, this review is distinct because the authors focus on the T- and D-arm interaction and related modified nucleosides.  Furthermore, the evolutionary process of tRNA is considered.  Therefore, I believe that this manuscript has scientific merits to be published in the journal, “Genes”.  The topics of this review match to the editorial policy of special issue, “Functions and Dynamics of RNA Modifications”.  However, I would like to recommend the authors some minor revisions before the publication.

Minor Points:

(1) Greek characters in the main text may be not correctly converted to the pdf. For example, Page 4, in the section “Modified nucleotides within the T-loop and G18 stabilize the elbow of tRNA”, the letters of pseudouridine (Ψ) are missing.  Please check through the manuscript.

(2) Abbreviations (m5s2U and m1s4Ψ). These abbreviations are not defined in the main text.  In several parts (for example, Figure 1), 5-methyl-2-thiouridine is abbreviated as “s2m5U”.  I think that “m5s2U” is better than “s2m5U”.  By the same reason, “m1s4Ψ” may be better than “s4m1Ψ”.  Please check through the manuscript and unify the abbreviations.

(3) Page 5 line 13: “This m1I can be …. 2’-O-ribose methylated derivative (m1Im) [23].”

In the reference [23], the presence of m1Im in tRNA mixtures from Thermoproteus, Pyrobabaculum and Pyrodictium was reported.  As far as I know, however, the position of m1Im has not been experimentally confirmed as position 57.  To avoid misunderstanding by general readers, the sentence should be changed.

(4) The position 54 in initiator tRNAMet from Sulfolobus acidocaldarius has been reported as Um54: Kuchino et al. (1982) Nature, 298, 684-685. Furthermore, the authors (Prof. Droogmans and Prof. Grosjean) recently reported the presence of s2U54 in several tRNAs from Sulfolobus acidocaldarius: Wolff et al. (2020) RNA 26, 1957-1975.  The information should be added to this review. 

(5) As the authors know, the complete sequence information of archaeal tRNAs is limited. If necessary, please cite these publications.

(i) Several tRNAs from Halobacterium salinarum: Gu et al. (1983) Nucleic Acids Res. 11, 5433-5442.

The sequence “m1Ψ54-Ψ55-Cm56-A57-A58” was found.

(ii) Several tRNAs from Haloferax volcanii: Gupta (1984) J. Biol. Chem. 259, 9461-9471.

The sequence “m1Ψ54-Ψ55-Cm56-A57-A58” was also found.

(iii) Elongator tRNAMet from Thermoplasma acidophilum: Kilpatrick et al. (1981) Nucleic Acids Res. 9, 4387-4390.

The sequence “Ψ54-Ψ55-Cm56-G57-A58” was found.

(iv) tRNALeu from Thermoplasma acidophilum: Tomikawa et al. (2013) FEBS Lett. 587, 3575-3580.

The sequence “Y54-Y55-Cm56-A57-m1A58” was found.

(v) tRNATrp from Thermococcus kodakarensis: Hirata et al. (2019) J. Bacteriol. 201: e00448-19.

The sequence “m5s2U54-Ψ55-Cm56-m1I57-m1A58” was experimentally confirmed.

(vi) tRNAIle from Haloarcula marismortui: Mandal et al. (2010) Proc. Natl. Acad. Sci. USA 107, 2872-2877.

The sequence “m1Ψ54-Ψ55-Cm56-m1I57-A58” was reported.

(6) In some parts, the order of reference is not good. Please check through the manuscript.  For example, Page 10 line 21, [47][67][63][103][72][104][105] should be [47][63][67][72][103][104][105]: Page 11 line 39, [122][123][47][124] should be [47][122] [123][124].

(7) Page 11 line 4: "THUMP is an RNA binding module that …. [116][117]."

These publications may be cited in this sentence.

Waterman et al. (2006) J. Mol. Biol. 356, 97-110.

Fislage et al. (2012) Nucleic Acids Res. 40, 5149-5161.

Neumann et al. (2014) Nucleic Acids Res. 42, 6673-6685.

Hirata et al. (2016) Nucleic Acids Res. 44, 6377-6390.

Author Response

Responses to referee #1:

We warmly thank the referee for constructive remarks and compliments.

(1) Greek characters in the main text may be not correctly converted to the pdf. For example, Page 4, in the section “Modified nucleotides within the T-loop and G18 stabilize the elbow of tRNA”, the letters of pseudouridine (Ψ) are missing. Please check through the manuscript.

All Greek characters have been restored. Our original text was in “Times New Roman” font characters while the Genes manuscript is in “Palatino Linotype”. This might be the reason why our initial Greek characters were lost after conversion.

(2) Abbreviations (m5s2U and m1s4Ψ). These abbreviations are not defined in the main text. In several parts (for example, Figure 1), 5-methyl-2-thiouridine is abbreviated as “s2m5U”. I think that “m5s2U” is better than “s2m5U”. By the same reason, “m1s4Ψ” may be better than “s4m1Ψ”. Please check through the manuscript and unify the abbreviations.

As recommended by the referee we now adopt the use of m5s2U/m1s4Psi all over the manuscript and in the figures.

(3) Page 5 line 13: “This m1I can be ….2’-O-ribose methylated derivative (m1Im) [23].” In the reference [23], the presence of m1Im in tRNA mixtures from Thermoproteus, Pyrobabaculum and Pyrodictium was reported. As far as I know, however, the position of m1Im has not been experimentally confirmed as position 57. To avoid misunderstanding by general readers, the sentence should be changed.

Indeed, information about m1Im came from analysis of bulk tRNA and therefore cannot be assigned to a given position in a single tRNA species. However, taking into account that m1I57 is a hallmark of archaeal tRNAs, it is highly probable that m1Im is found at position 57. Therefore we maintain the m1Im but now as (m1Im57) (in brackets and italics) in all figures with corresponding explanation in the legend of figure4 and in the text. The new sentence in the text is:

Page 4; line 31

2’-O-ribose methylation of cytidine at position 56 (Cm) was found in tRNAs of the crenarchaeon Pyrobaculum aerophilum [24] and also of N1-methyl-inosine at position 57 (m1Im) in tRNAs of Pyrobaculum islandicum and of Pyrodictium occultum. However, in this latter case information comes from analysis of bulk tRNA with no evidence for the presence of m1Im at position 57 exclusively [25].

In the legend of figures 1 and 4 we have added:

Page 3; line 9 and Page 6; line 8

2’-O-methylation of m1I57 into m1Im57 is putative (see text).

(4) The position 54 in initiator tRNAMet from Sulfolobus acidocaldarius has been reported as Um54: Kuchino et al. (1982) Nature, 298, 684-685. Furthermore, the authors (Prof. Droogmans and Prof. Grosjean) recently reported the presence of s2U54 in several tRNAs from Sulfolobus acidocaldarius: Wolff et al. (2020) RNA 26, 1957-1975. The information should be added to this review.

In a short paper in Nature (without much details in Materials and Methods section) Kuchino et al (1982) indeed reported the presence of Um54 in initiator tRNA of S. acidocaldarius (the only sequenced tRNA of this organism). This however has never been confirmed by other groups. In order not to elude the problem we propose the following remark at the end of the paragraph related to U54 modifications:

Page 11; line 23

Finally, in the case of the hyperthermophilic crenarchaeon Sulfolobus acidocaldarius, s2U has been found at position 54 of several tRNAs. In the particular case of tRNAVal, a methylthio U-derivative was also found, but its chemical structure remains elusive [29]. Examination of the S. acidocaldarius genome reveals the presence of genes coding for TtuA and TtuB responsible for thiolation of U54 but no gene coding for TrmA (catalyzing m5U54), Pus10 (catalyzing Ψ54 in addition of Ψ55) nor TrmY (catalyzing m1Ψ54) was found. One possibility is that ribose methylation of U54 occurs in only a few tRNA species, such as the initiator tRNA [129]. Indeed, 2’-O-methylated nucleotides are found at several positions of various S. acidocaldarius tRNAs and are known to be formed by the guide RNA- dependent FlpA machinery [130][131]. However, no guide RNA specific for U54 methylation has been reported up to now. Therefore, in absence of a clearcut evidence for Um54 existence and of convincing information about the corresponding methylation system, this problem remains unsolved.

We have also completed the information relative to s2U54 in figure 5.

(5) As the authors know, the complete sequence information of archaeal tRNAs is limited. If necessary, please cite these publications:

(i) Several tRNAs from Halobacterium salinarum: Gu et al. (1983) Nucleic Acids Res. 11, 5433-5442. The sequence “m1Ψ54-Ψ55-Cm56-A57-A58” was found.

(ii) Several tRNAs from Haloferax volcanii: Gupta (1984) J. Biol. Chem. 259, 9461-9471.

The sequence “m1Ψ54-Ψ55-Cm56-A57-A58” was also found.

(iii) Elongator tRNAMet from Thermoplasma acidophilum: Kilpatrick et al. (1981) Nucleic Acids Res. 9, 4387-4390. The sequence “Ψ54-Ψ55-Cm56-G57-A58” was found.

(iv) tRNALeu from Thermoplasma acidophilum: Tomikawa et al. (2013) FEBS Lett. 587, 3575-3580. The sequence “Y54-Y55-Cm56-A57-m1A58” was found.

(v) tRNATrp from Thermococcus kodakarensis: Hirata et al. (2019) J. Bacteriol. 201: e00448-19. The sequence “m5s2U54-Ψ55-Cm56-m1I57-m1A58” was experimentally confirmed.

(vi) tRNAIle from Haloarcula marismortui: Mandal et al. (2010) Proc. Natl. Acad. Sci. USA 107, 2872-2877. The sequence “m1Ψ54-Ψ55-Cm56-m1I57-A58” was reported.

Most references recommended by the referee concern single tRNA species in given organisms. Therefore, to be more general, at page 5 instead adding all selected cases mentioned by the referee, we prefer to add in the manuscript:

Page 5; line 17

Notice, the presence of m1A58 and/or m1I57 depends much on the archaeon considered and the type of tRNA analyzed (especially the nature of nucleotide-59, see below). For example, most tRNAs from halophilic Archaea such as Haloferax volcanii display m1I57 and unmodified A58 [27][28], while tRNAs of thermophilic and hyperthermophilic Archaea often display m1A58 and/or m1I57 [29] and references therein.

(6) In some parts, the order of reference is not good. Please check through the manuscript.  For example, Page 10 line 21, [47][67][63][103][72][104][105] should be [47][63][67][72][103][104][105]: Page 11 line 39, [122][123][47][124] should be [47][122] [123][124].

As requested by the referee, all cited references have been ordered by increasing numbers in the whole manuscript.

 (7) Page 11 line 4: "THUMP is an RNA binding module that …. [116][117]."

These publications may be cited in this sentence.

Waterman et al. (2006) J. Mol. Biol. 356, 97-110.

Fislage et al. (2012) Nucleic Acids Res. 40, 5149-5161.

Neumann et al. (2014) Nucleic Acids Res. 42, 6673-6685.

Hirata et al. (2016) Nucleic Acids Res. 44, 6377-6390.

Indeed presence of THUMP motif has been discussed in many papers. We have mentioned only the first historical one by Aravind et al (2001) and only one example among many others. However, because they illustrate well the importance of the tRNA core for binding the modification enzymes acting at positions 6, 8 and 10 of tRNA, we now extend the citations to the four additional ones recommended above by the referee.

Reviewer 2 Report

In this review the authors summarize the current knowledge about the T-loop of tRNAs in different domains of life and the known chemical modifications of its highly conserved RNA bases. They conclude a large number of literature and provide an interesting and very focused perspective. It was an interesting and enjoyable read. I am sure this review will be recognized in the field and provide a very useful resource for newcomers and experts.

I have only two minor suggestions/issues and I strongly recommend to accept this manuscript for publication in “Genes”.

  • Recently there were several reviews about the cross talk and internal circuitry between tRNA modifications and their modification enzymes (see e.g. Han and Phizicky 2018, Sokołowski et al 2018, Dixit et al 2018). Maybe the authors could add a sentence about the sequential order and the possibility of interdependency between individual T-loop modifications
  • In my version of the text most of the “ψ” appear as empty space – please double check.

Author Response

Responses to referee #2:

We warmly thank the referee for constructive remarks and compliments.

(1) Recently there were several reviews about the cross talk and internal circuitry between tRNA modifications and their modification enzymes (see e.g. Han and Phizicky 2018, Sokołowski et al 2018, Dixit et al 2018). Maybe the authors could add a sentence about the sequential order and the possibility of interdependency between individual T-loop modifications.

Indeed, cross talk/internal circuiting between tRNA modifications and their modification enzymes is an important new concept that concerns the whole tRNA and not only the limited T-loop region. The following sentence has therefore been added at the end of the paragraph:

Page 12; line 11

Interdependency between different nucleotide modifications and their modification pathways has been found to be particularly important for modifications in the anticodon-stem loop [60,140] [141].

(2) In my version of the text most of the “ψ” appear as empty space – please double check.

All Greek characters have been restored. Our original text was in “Times New Roman” font characters while the Genes manuscript is in “Palatino Linotype”. This might be the reason why our initial Greek characters were lost after conversion.